# The Impairment of Cell Metabolism by Cardiovascular Toxicity of Doxorubicin Is Reversed by Bergamot Polyphenolic Fraction Treatment in Endothelial Cells

**DOI:** 10.3390/ijms23168977

**Published:** 2022-08-11

**Authors:** Cristina Algieri, Chiara Bernardini, Francesca Oppedisano, Debora La Mantia, Fabiana Trombetti, Ernesto Palma, Monica Forni, Vincenzo Mollace, Giovanni Romeo, Ilaria Troisio, Salvatore Nesci

**Affiliations:** 1Department of Veterinary Medical Sciences, University of Bologna, 40064 Ozzano dell’Emilia, Italy; 2Department of Health Sciences, Institute of Research for Food Safety & Health (IRC-FSH), University “Magna Graecia” of Catanzaro, 88100 Catanzaro, Italy; 3Health Sciences and Technologies-Interdepartmental Center for Industrial Research (CIRI-SDV), Alma Mater Studiorum, University of Bologna, 40126 Bologna, Italy; 4Medical Genetics Unit, Sant’Orsola-Malpighi University Hospital, 40126 Bologna, Italy

**Keywords:** cell metabolism, bergamot polyphenolic fraction, doxorubicin, mitochondria, porcine aortica endothelial cells

## Abstract

The beneficial effects of bergamot polyphenolic fraction (BPF) on the mitochondrial bioenergetics of porcine aortic endothelial cells (pAECs) were verified under the cardiotoxic action of doxorubicin (DOX). The cell viability of pAECs treated for 24 h with different concentrations of DOX was reduced by 50%, but the negative effect of DOX was reversed in the presence of increasing doses of BPF (100 µg/mL and 200 µg/mL BPF). An analysis of the protective effect of BPF on the toxic action of DOX was also carried out on cell respiration. We observed the inhibition of the mitochondrial activity at 10 µM DOX, which was not restored by 200 µg/mL BPF. Conversely, the decrease in basal respiration and ATP production caused by 0.5 or 1.0 µM DOX were improved in the presence of 100 or 200 µg/mL BPF, respectively. After 24 h of cell recovery with 100 µg/mL or 200 µg/mL BPF on pAECs treated with 0.5 µM or 1.0 µM DOX, respectively, the mitochondrial parameters of oxidative metabolism impaired by DOX were re-boosted.

## 1. Introduction

It has been ascertained that features of bergamot polyphenolic fraction (BPF) boost the bioenergetic parameters of mitochondria, improve cellular oxidative metabolism and block molecular events in mitochondria that trigger different forms of regulated cell death [1].

Doxorubicin (DOX) is an apoptosis-inducing anticancer agent; in particular, it is a cytotoxic agent that stimulates ROS production and inhibits DNA and RNA synthesis [2,3]. It is known that anticancer agents can cause damage to endothelial cells by indirectly reducing the normal functioning of the circulatory system, causing myocardial toxicity and congestive heart failure [4]. Furthermore, DOX-induced cardiomyocyte dysfunction is linked to the altered use of the energy substrate [5]. DOX is the most widely used anthracycline in numerous neoplastic diseases, despite the fact that its cardiotoxicity is an important side effect [5,6]. In fact, in addition to causing acute toxicity that is reversible, DOX, over the years, can generate a progressive injury process, causing a dose-dependent risk of heart failure [5,6,7]. At the level of tumor cells, DOX inhibits proliferation and leads to cell death as it intercalates with DNA in the nucleus and inhibits topoisomerase IIα (Top IIα) [5]. On the other hand, Top IIβ is strongly expressed in adult cardiomyocytes, both in the nucleus and in the mitochondria. DOX can be conjugated with Top IIβ and intercalated with both nuclear DNA and mitochondrial DNA, activating the DNA damage response and apoptosis [8,9]. DOX-induced endothelial dysfunction is critical in the progression of cardiovascular disease in patients treated with this anthracycline [10].

The most frequently described molecular process linked to DOX-induced cardiotoxicity is oxidative stress [11]. Oxidative stress induced by DOX determines the onset of redox modifications at the level of biological macromolecules [12,13]. Mitochondria, occupying up to 35% of total cardiomyocyte volume, are the main organelles damaged in the heart by DOX. In particular, DOX interacts with cardiolipin, which represents a specific phospholipid of the inner mitochondrial membrane and participates in the stabilization of oxidative phosphorylation (OXPHOS) complexes [5]. In particular, the accumulation of DOX in the mitochondria determines the uncoupling of OXPHOS complexes, with a consequent reduction in adenosine triphosphate (ATP) synthesis and a simultaneous increase in the AMP/ATP ratio, resulting in the activation of AMP-activated protein kinase. DOX-induced mitochondrial dysfunction also alters the intracellular calcium balance [14,15]. DOX has been shown to reduce mitochondrial membrane potential, open mitochondrial permeability transition pore (mPTP), and activate the ROS-induced ROS release mechanism, resulting in mitochondrial dysfunction, causing damage to the vascular endothelium (VE) [16]. In fact, DOX also damages non-cardiomyocyte heart cells such as endothelial cells (ECs) [5,11,14]. DOX can enter ECs through SLC22A1 and interfere, for example, with endothelial nitric oxide synthase (eNOS), which is the NOS isoform most sensitive to DOX [12,17].

The Mediterranean diet (MD), characterized by lower consumption of red meat and a prevalent intake of olive oil and plant-based foods, shows a positive correlation with the reduced onset and progression of some diseases, such as cancer, diabetes mellitus and coronary heart disease [18]. A particularly interesting role can be attributed to bergamot (*Citrus bergamia Risso et Poiteau*), a plant that grows exclusively in the southernmost area of the Calabrian Ionian coast (Southern Italy) [18,19]. The soil and the microclimate of this geographical area are decisive for the organoleptic characteristics of bergamot, in particular for its high concentration of polyphenols [18,19]. As regards antioxidant and anti-inflammatory action and the reduction of hyperlipidemia and hyperglycemia, these are attributable to BPF, which represents the polyphenol-rich fraction obtained from the juice and albedo of bergamot [1]. In recent years, researchers have been looking for treatments that can reduce the cardiotoxicity of DOX [20]. The use of natural compounds [21], including BPF [15], is particularly interesting. In adult male Wistar rats with DOX-induced heart damage, Carresi et al. monitored the antioxidant and cardioprotective effect of BPF. In addition, the same authors also tested the effects of BPF on cardiomyocyte survival and resident endogenous cardiac stem/progenitor cell activation [22]. The study showed that after 21 days of treatment with DOX, cardiac function was reduced and recovery was reported in the presence of BPF [22]. In particular, in a recently published paper, in experiments with swine heart mitochondria, we evaluated the effect of BPF on F_1_F_O_-ATPase activity. The data obtained showed that BPF positively influenced the activity of the enzyme-dependent Mg^2+^ cofactor, while inhibiting the activity of Ca^2+^-dependent F_1_F_O_-ATPase. The mPTP phenomenon is triggered by Ca^2+^-activated F_1_F_O_-ATPase [23]. Furthermore, in the presence of BPF, we observed a reduction in the opening of mPTP [1]. The study also demonstrated the effect of BPF on cellular metabolism and demonstrated an effect on ATP production in porcine aortic endothelial cells (pAECs) [1]. Notably, BPF did not affect total intracellular ATP levels, but increased ATP production was observed through oxidative metabolism rather than through glycolysis. Cell viability also increased in the presence of BPF. Therefore, that study made it possible to obtain an innovative result relating to the positive effect of BPF on mitochondrial bioenergetics and oxidative metabolism [1].

The aim of the present study was to determine the effect of BPF on pAECs in physiological and stress conditions induced by DOX, as a function of the known endothelial vascular toxicity of the chemotherapeutic drug. To summarize, BPF could potentially have a beneficial effect on the vascular endothelium by improving mitochondrial function and avoiding the cardiotoxic action of DOX.

## 2. Results

Different DOX concentrations negatively affected the physiology of pAECs, impairing the cellular metabolism. The toxic effect was characterized by decreased mitochondrial energy efficiency. The BPF treatments of pAECs, overcoming the action of DOX on bioenergetics, promoted the improvement of mitochondrial features in ATP production.

### 2.1. Toxic Action of DOX on the Viability of pAECs and the Effects of BPF

The protective effect of BPF against the toxic effects of DOX was investigated. pAECs exposed to DOX lost their typical morphology of a continuous monolayer and an increasing number of detached cells appeared at all doses tested (Figure 1A–D); BPF reduced DOX’s toxic effect (Figure 1E). A decrease of about 50% in cell viability was induced at all the DOX concentrations tested. Independently of the BPF concentration in DOX-treated pAECs, the cell viability reached the 70% threshold; therefore, BPF significantly reduced the DOX-induced cytotoxicity at both doses studied. The presence of 0.5 µM DOX plus 100 µg/mL BPF was the only condition in which the toxicity of DOX was reversed, and there was no statistically significant difference between control versus DOX with BPF (Figure 1F).

### 2.2. Protection from Damage of DOX by BPF on Mitochondrial Metabolism of the pAECs

We compared the remodeling of pAEC metabolism in the presence of DOX with and without BPF. We took two different approaches, one with a simultaneous treatment (DOX + BPF) for 24 h (Figure 2A,B, Figure 3A,B and Figure 4A,B) and another with the recovery of the DOX group of cells over another 24 h period with only BPF treatment (Figure 3C,D and Figure 4C,D). OCR values were obtained using serially injected mitochondrial modulators of respiration (MMR), i.e., olig, FCCP and Rot+AA. In pAECs, the toxic effect of 10 µM DOX on mitochondrial respiration was not restored by 200 µg/mL BPF. These results are highlighted by the linear cell respiration profile observed in the presence of DOX, which was insensitive to MMR compared to cells in the absence of DOX or with only 200 µg/mL BPF (Figure 2A). Accordingly, all the bioenergetic parameters were inhibited in the presence of DOX or DOX + BPF (Figure 2B). Moreover, we observed a time- and concentration-dependent reduction of pAECs bioenergetic parameters in presence of 200 μg/mL BPF at 24 h (Figure 2 and Figure 3A,B) and a little more accentuated at 48 h (Figure 3C,D). However, the MTT test was not affected by the different exposure of pAECs to 100 μg/mL or 200 μg/mL BPF (data not shown).

Conversely, we noticed differences in cell respiration profiles in the presence of 1 µM DOX with and without 200 µg/mL BPF (Figure 3A,C) or in the presence of 0.5 µM DOX with and without 100 µg/mL BPF (Figure 4A,C). In the presence of 1 µM DOX, the impaired bioenergetic parameters of pAECs showed a significant change in the OCR values of basal, proton leak, maximal respiration and ATP production, with decreases in their activities of about 55%, 40%, 42% and 63%, respectively (Figure 3B). The toxic effect of DOX on mitochondrial respiration was preserved in pAECs co-treated with 200 µg/mL BPF. Indeed, the significant increases of 37%, 35%, 20% and 66% OCR observed in the above parameters occurred due to the action of BPF.

Very interesting results are reported in Figure 3C,D. Recovery treatment BPF for 24 h restored the DOX-altered mitochondrial parameters on pAECs. In the presence of 200 µg/mL BPF, basal, proton leak, maximal respiration and ATP production retrieved the cell respiration and the OCR activity of these mitochondrial parameters if compared to DOX-treatment. No statistically significant differences were recorded between BPF and DOX + BPF, suggesting the dismission of the DOX’s toxic effect on pAECs (Figure 3D). Importantly, there was no restoration of mitochondrial parameters in DOX-treated cells when BPF was not added (Figure 3D).

However, the dose-dependent and time-dependent DOX toxicity was reverted by the effect of BPF on pAECs’ metabolism. This trend was also observed with 0.5 µM DOX and 100 µg/mL BPF. Indeed, the results shown in Figure 4 corroborated the protective effect of BPF on the toxic action of DOX. Moreover, pAECs treated with 100 µg/mL BPF increased the OCR by 20% of basal and 15% of ATP production compared with controls. The inhibition of basal, proton leak, maximal respiration and ATP production by 0.5 µM DOX on mitochondrial respiration was cancelled by simultaneous treatment with BPF. We only noticed a decrease in the spare respiratory capacity in all the conditions compared with the controls (Figure 4A,B). The recovery highlighted that the cell respiration profile in pAECs treated and untreated with BPF were the same. Moreover, pAECs could attenuate the toxic effect of DOX on OCR values after 24 h of interruption of drug exposure (Figure 4C). Indeed, the effect of a low DOX concentration was reverted by simultaneous treatment with BPF by increasing the mitochondrial parameters at OCR values that were not significantly different from the control values (Figure 4D).

## 3. Discussion

DOX treatment in chemotherapy medication has a high cardiotoxicity risk and the adverse side effects invalidate its clinical use [6]. However, in a broad group of cancers, including leukemias, lymphomas and several solid tumors, namely, breast, gynecological, urogenital, endocrine, brain tumors and stomach cancer, as well as Ewing and Kaposi’s sarcomas, anthracyclines are the standard chemotherapy backbone and DOX is the cornerstone of chemotherapy [24]. The arising cardiovascular toxicity can trigger the development of heart failure. In particular, endothelial cells undergo an environmental insult with the dysregulation of inflammatory and vascular reparative functions, leading to endothelial cell death and the progression of cardiomyopathy [10]. Despite this, DOX is widely used to treat cancer, which is the most prominent cause of death globally, even if, as a side effect, it is the cause of cardiovascular diseases, which have among the highest mortality rates in the world.

DOX causes mitochondrial dysfunction, acting on OXPHOS with an oxidative stress insult, culminating in the alteration of gene expression. In mitochondria, DOX exhibits redox cycling, interacting with the complex I or cytochrome P450 system, and it changes into a semiquinone radical that, in the presence of molecular oxygen, is rapidly re-oxidized and forms an anion superoxide. The damage to membrane lipids and/or proteins (in particular, thiol groups) is thus the result of the conversion of free radicals of oxygen to other ROS. Moreover, the DOX semiquinone radical might also undergo a second molecular transformation, responsible for DNA modification by alkylation, or may favor its insertion between the DNA planar bases via intercalation [25].

Understanding the molecular mechanism underlying mitochondrial dysfunction in DOX-induced endothelial dysregulation is essential in order to reduce the limitations involved in the necessary use of DOX in anticancer chemotherapy. DOX-dependent endothelial dysfunction has been ascertained in our experiments in the observation of the halving of the viability of DOX-treated pAECs (Figure 1). The toxicity of DOX was highlighted in the results obtained from cell metabolism. In particular, mitochondrial respiration was negatively affected in the presence of DOX. The reductions observed in all the bioenergetic parameters corroborated the inhibitory effect on OXPHOS of pAECs (Figure 2, Figure 3 and Figure 4). However, the known beneficial effects of BPF on mitochondrial bioenergetics and oxidative metabolism [1] were confirmed to prevent the impaired mitochondrial function of DOX-treated pAECs. Interesting, BPF might have a quenching action on mitochondrial dysfunction induced by DOX when administered simultaneously with the drug and, importantly, BPF can obtain a therapeutic effect by improving the mitochondrial parameters after DOX treatment, increasing the cell metabolism at the level of pAECs. Conversely, pAECs that suffered a DOX insult were unable to restore the performance of the activity of mitochondrial oxidative metabolism. These effects were more evident at low concentrations of DOX and BPF. However, the oxidative metabolism of pAECs with 10 µM DOX treatment was not rescued by BPF. The DOX-dependent disordering of mitochondrial respiration becomes irreversible, in particular, in cases where the phenomenon of mPTP might occur.

DOX-related oxidative damage in cardiovascular tissue is dependent on ROS generation rates, which triggers mPTP. With the opening of mPTP, a decrease in the electrochemical gradients across the inner mitochondrial membranes due to its high channel conductivity leads to the avoidance of mitochondrial ATP production, which triggers different forms of regulated cell death [26]. The mPTP phenomenon is calcium- and redox-regulated [27] by means of pro-oxidant agents such as DOX. The protective effect of BPF on the cell metabolism of pAECs treated with DOX might help to evade the redox stress response to DOX toxicity in mitochondria. Therefore, the antioxidant protection of BPF could have an important role in improving mitochondrial bioenergetics.

## 4. Materials and Methods

### 4.1. Chemicals and Reagents

BPF was purchased from the Herbal & Antioxidant Derivatives (H&AD) company (Bianco, Italy). Doxorubicin was obtained from Sigma-Aldrich (Milan, Italy). Seahorse XF Assay Kits and Reagents were purchased from Agilent, Santa Clara, CA, USA. Quartz double-distilled water was used for all reagent solutions.

### 4.2. Preparation of the Bergamot Polyphenolic Fraction (BPF)

*Citrus bergamia Risso & Poiteau* fruits were collected along the Ionian coast of Calabria, between Bianco and Reggio Calabria, Italy. Bergamot juice was obtained after peeling and squeezing the fruit. This phase was followed by the stripping process, in order to deplete the juice from the oil fraction, and the ultrafiltration process was used to clarify it. The juice thus obtained was made to elute through a polystyrene resin column in order to retain the polyphenolic compounds having molecular weights between 300 and 600 Da. A KOH solution was used to elute the polyphenolic fractions. In order to reduce the furocoumarin content, the basic eluate was incubated on a rocking platform, and the duration of the shaking phase was established as a function of the amount of furocoumarin contaminants. Subsequently, this phytocomplex was neutralized via filtration on cationic resin at an acidic pH, dried under vacuum and minced to obtain BPF powder. Finally, the BPF powder was analyzed to determine the content of flavonoids, furocoumarins and other polyphenols, and by means of UHPLC-HRMS/MS, it was possible to observe that BPF consisted of 40% flavonoids, whereas 60% included carbohydrates, fatty acids, pectins and maltodextrins. High-resolution mass spectrometry (Orbitrap spectrometer) and HRMSMS (ddMS2, data-dependent MS/MS) were used to define the flavonoid profile in neoeriocitrin, naringin and neohesperidin, as well as in the whole HMG family, including bruteridin and melitidin, and flavonoids such as 6.8-di-C-glycosides [1,22,28].

### 4.3. Cell Cultures

Primary cell cultures of pAECs were isolated and maintained as previously described [29]. pAECs from 3 to 8 passages (P) were used to perform the experiments. The cells were seeded and routinely cultured in T25 or T75 primary culture flasks (2 × 10^4^ cells/cm^2^) in a human endothelial serum-free medium (hESFM), added to a 5% Fetal Bovine Serum (FBS) and 1× antibiotic/antimycotic solution in a 5% CO_2_ atmosphere and at 38.5 °C. An inverted Eclipse Microscope (TS100) with a digital C-Mount Nikon photo camera (TP3100) was used to define the cell morphology.

### 4.4. Cell Viability

To evaluate the protective effect of BPF on the toxic effect of DOX treatment, pAECs were treated with 0.5, 1 and 10 µM DOX in the presence or absence of BPF (100 or 200 µg/mL) for 24 h, and then fresh medium with or without BPF was changed and incubated for an additional 24 h. Viability was determined using the MTT test. After adding the culture medium with the MTT substrate, a three-hour incubation was performed. The MTT solubilization solution allowed the dissolution of the formazan crystals. The Infinite^®^ F50/Robotic absorption microplate readers from TECAN Life Sciences (Männedorf, Switzerland) made it possible to measure the absorbance of formazan (Abs) at a wavelength of 570 nm; hence, the background absorbance was subtracted from the multiwell plates, measured at 690 nm.

### 4.5. Cellular Metabolism

Using the Seahorse XFp analyzer (Agilent, Santa Clara, CA, USA), studies of cellular energy metabolism were carried out by measuring the oxygen consumption rate (OCR) and the cellular respiration index (pmol/min), as well as the rate of extracellular acidification (ECAR) and the glycolysis index (mpH/min).

The pAECs (20 × 10^3^/well) were seeded in XFp cell culture mini-plates (Agilent, Santa Clara, CA, USA). The day after pAECs were treated with 0.5, 1 and 10 µM DOX in the presence or absence of BPF (100 or 200 µg/mL) for 24 h, then fresh medium with or without BPF (100 or 200 µg/mL) was changed and incubated for an additional 24 h. At the end of the treatment, the culture medium was replaced with Seahorse XF DMEM medium, pH 7.4, supplemented with 10 mM glucose, 2 mM L-glutamine and 1 mM sodium pyruvate. The analyses were conducted in the absence of BPF (control), in the presence of BPF and with DOX with and without BPF for the Mito Stress Test. OCR and ECAR were measured with the Cell Mito Stress Test program for 45 min at 37 °C. In addition, the injection ports of the XFp sensor cartridges were hydrated overnight with the XF calibrant at 37 °C. On the day of analysis, the cartridges were loaded with 10 times the concentration of inhibitors, as indicated in the instructions for the Cell Mito Stress Test. Final concentrations were 1.5 μM oligomycin (olig) (port A), 1.0 μM carbonyl cyanide-4-(trifluoromethoxy) phenylhydrazone (FCCP) (port B) and 0.5 μM rotenone plus antimycin A (port C) [1].

Using the Mito Stress Test, it was possible to obtain information on cellular respiration through the following parameters: basal respiration, the basic OCR detected before the addition of oligomycin; minimal respiration, measured via the OCR in the presence of oligomycin; and maximum respiration, the OCR after the addition of FCCP, as well as the proton leak, which corresponds to the difference between basal respiration and respiration in the presence of oligomycin (minimal respiration), and non-mitochondrial respiration, evaluated in the presence of rotenone plus antimycin A (respiratory chain inhibitors). The latter was subtracted from all the above parameters. ATP production was obtained by assessing the difference between basal respiration and minimal respiration (OCR in the presence of oligomycin), whereas the difference between maximal and basal respiration was used to determine the spare respiratory capacity. The analysis was performed at 37 °C. The parameter values, analyzed using WAVE software, were calculated per well, according to the manufacturer’s instructions, on three independent experiments and were normalized to the total number of cells per well [30]. Moreover, the WAVE software has been set to normalize the results based on the pAECs’ vitality obtained from the MTT test.

### 4.6. Statistical Analyses

Statistical analyses were performed using SIGMASTAT software. Each treatment was replicated three or eight times (viability test) in three independent experiments. Data were analyzed using Student’s *t*-test, or via a one-way analysis of variance (ANOVA) followed by the Student–Newman–Keuls test when F values indicated significance (*p* ≤ 0.05).

## 5. Conclusions

We have determined that the metabolic alterations in pAECs caused by DOX can be reversed with the use of BPF. Therefore, these results demonstrate the protective effect of BPF in cardiovascular pathologies and even highlight, at the molecular level, how the toxic action of DOX, perhaps explained by its redox-remodeling mechanism, is reversed by the known antioxidant powers of BPF. Restoring the correct metabolic cellular functions can avoid alterations affecting the VE and can act positively on the cardiovascular disorders resulting from the toxic action of DOX.

## Figures and Tables

**Figure 1 ijms-23-08977-f001:**
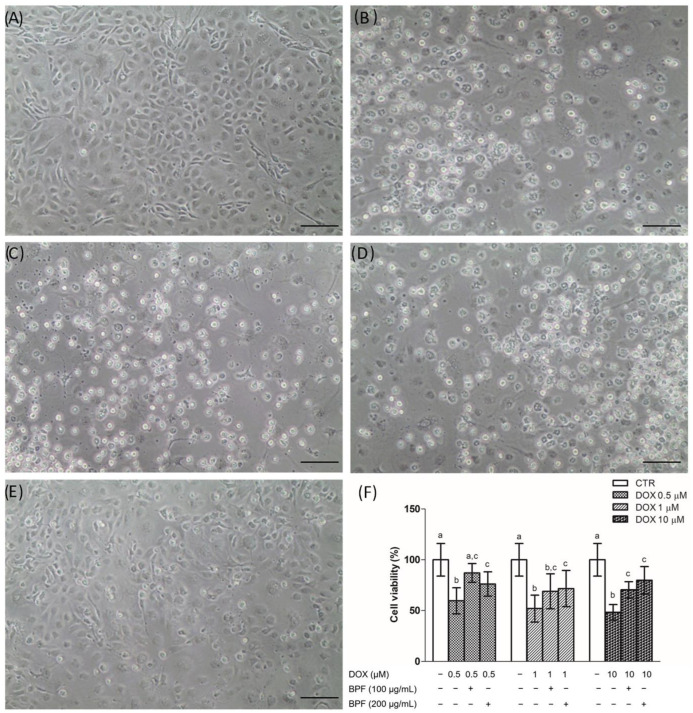
Effect of BPF on DOX-induced pAEC cytotoxicity. (**A**–**E**) Representative images of pAECs treated with DOX 0.5, 1, 10 µM (**B**,**C**,**D**, respectively) in the absence or presence of 100 µg/mL BPF (**E**). (**F**) Cell viability quantification: each bar represents the mean ± SD of three independent experiments. Scale bar (―) 100 µm. Different letters above the bars indicate significant differences (*p* < 0.05, one-way ANOVA, post hoc Tuckey comparison test) between each DOX treatment with and without BPF and the control (CTR) group.

**Figure 2 ijms-23-08977-f002:**
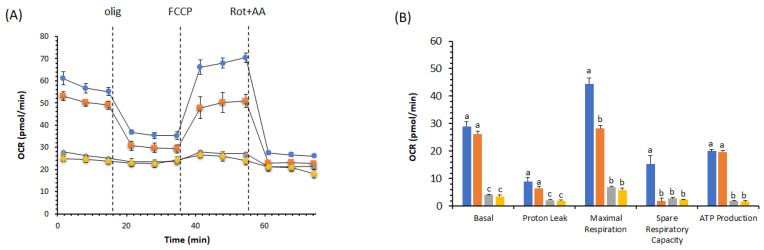
Effect of 200 µg/mL BPF on the cell metabolism of pAECs treated with 10 µM DOX. The mitochondrial respiration profile was obtained from the oxygen consumption rate (OCR) in control (●, blue), 200 µg/mL BPF (■, orange), 10 µM DOX (▲, gray) and DOX plus BPF (♦, gold) treatment under basal respiration conditions and after the addition of 1.5 μM oligomycin (olig), 1.0 μM FCCP and a mixture of 0.5 μM rotenone plus antimycin A (Rot + AA). Inhibitor injections are shown as dotted lines (**A**). Mitochondrial parameters (basal respiration, proton leak, maximal respiration, spare respiratory capacity, ATP production) in control (█, blue), 200 µg/mL BPF (█, orange), 10 µM DOX (█, gray) and DOX plus BPF (█, gold) samples (**B**). Data expressed as points (**A**) and column charts (**B**) represent the means ± SD (vertical bars) from three experiments carried out on different cell preparations. Different letters indicate significant differences (*p* ≤ 0.05) among treatments within the same bioenergetic parameters.

**Figure 3 ijms-23-08977-f003:**
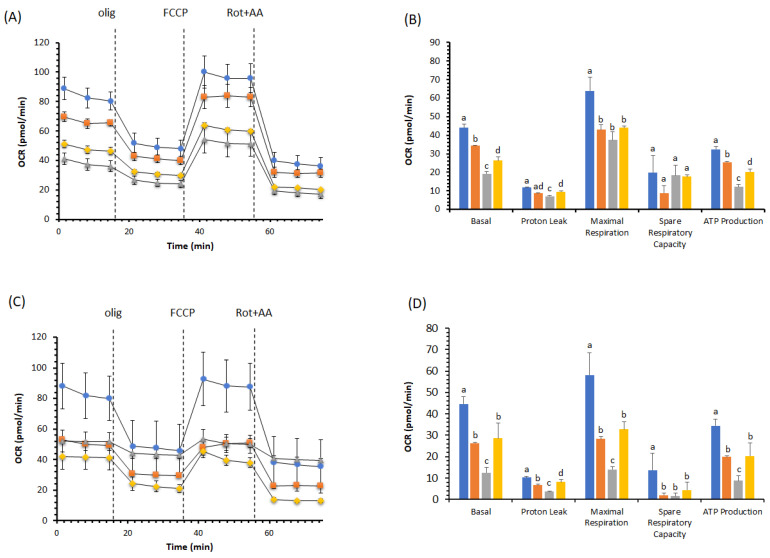
Effect of 200 µg/mL BPF on the cell metabolism of pAECs treated with 1 µM DOX. The action of DOX on mitochondrial respiration without (**A,B**) and with 24 h of recovery (**C,D**) in the presence of BPF. The mitochondrial respiration profile was obtained from the oxygen consumption rate (OCR) in control (●, blue), 200 µg/mL BPF (■, orange), 1 µM DOX (▲, gray) and 1 µM DOX plus BPF (♦, gold) treatments under basal respiration conditions and after the addition of 1.5 μM oligomycin (olig), 1.0 μM FCCP and a mixture of 0.5 μM rotenone plus antimycin A (Rot + AA). Inhibitor injections are shown as dotted lines (**A**,**C**). Mitochondrial parameters (basal respiration, proton leak, maximal respiration, spare respiratory capacity, ATP production) in control (█, blue), 200 µg/mL BPF (█, orange), 1 µM DOX (█, gray), and 1 µM DOX plus BPF (█, gold) samples (**B**,**D**). Data expressed as points (**A**,**C**) and column charts (**B**,**D**) represent the means ± SD (vertical bars) from three experiments carried out on different cell preparations. Different letters indicate significant differences (*p* ≤ 0.05) among treatments within the same bioenergetic parameters.

**Figure 4 ijms-23-08977-f004:**
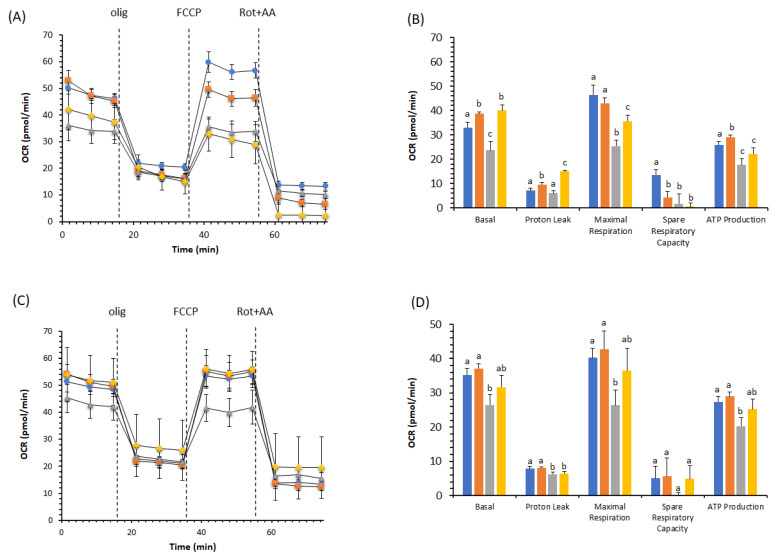
Effect of 100 µg/mL BPF on cell metabolism of pAECs treated with 0.5 µM DOX. The action of DOX on mitochondrial respiration without (**A,B**) and with 24 h of recovery (**C,D**) in the presence of BPF. The mitochondrial respiration profile was obtained from the oxygen consumption rate (OCR) in control (●, blue), 100 µg/mL BPF (■, orange), 0.5 µM DOX (▲, gray), and DOX plus BPF (♦, gold) treatments under basal respiration conditions and after the addition of 1.5 μM oligomycin (olig), 1.0 μM FCCP and a mixture of 0.5 μM rotenone plus antimycin A (Rot + AA). Inhibitor injections are shown as dotted lines (**A**,**C**). Mitochondrial parameters (basal respiration, proton leak, maximal respiration, spare respiratory capacity, and ATP production) in control (█, blue), 100 µg/mL BPF (█, orange), 0.5 µM DOX (█, gray) and DOX plus BPF (█, gold) samples (**B**,**D**). Data expressed as points (**A**,**C**) and column charts (**B**,**D**) represent the means ± SD (vertical bars) from three experiments carried out on different cell preparations. Different letters indicate significant differences (*p* ≤ 0.05) among treatments within the same bioenergetic parameters.

## Data Availability

Not applicable.

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
