# Peer review of "The Impairment of Cell Metabolism by Cardiovascular Toxicity of Doxorubicin Is Reversed by Bergamot Polyphenolic Fraction Treatment in Endothelial Cells"

_ijms, 2022, doi:10.3390/ijms23168977_

Round 1
Reviewer 1 Report
In this work, authors try to demonstrate a protective effect of bergamot polyphenolic fraction (BPF) from the cardiotoxic side-effects of doxorubricin (Dox) on porcine aortic endothelial cells.
They investigate the morphology and the bioenergetic parameters of this cells exposed or not to Dox and BPF.
Overall, while the manuscript is generally well written and quite clear, however there are some concerns that prevent its publication in the current form.
The main issue is that 200uM BPF appears to have an impact on cellular metabolism alone, as they inhibits both maximal and spare respiratory capacity in figure 2a-b, as well as basal, maximal and "ATP turnover" in figure 3a-b and 3c-d. The latter figure (3c) shows a rather alarming profile, with a curve for BPF alone that is more similar to the Dox one than to the control. This data seems to suggest a possible detrimental effect rather than a protective one. Indeed, I'd like to see the results on cell viability of BPF alone, as figure 1F shows only data in combination with Dox.
Conversely, a better adherence to authors' hypothesis is shown by 100uM BPF (fig 4), in which the protective effects appears to be more convincing. Did Authors investigated the effects of lower concentrations, such as 50uM BPF?
In the introduction and discussion sections, authors state that Dox exerts its action thorugh ROS production. Since ROS are easily measurable with flow cytometry, Authors should consider to include these experiments to strenghten their results and demonstrate that BPF causes ROS reduction.
Moreover, it is unclear how they performed normalization for the XF analyzer data, cell were counted at the seeding stage, before or after the analysis? Please provide an extended explanation, as if performed at the seeding stage with the 50% of dox mortality it would invalidate the results
minor points
- In the abstract, line 24-26, authors would mean that mitochondrial parameter are impaired or improved? Please reformulate the phrase in a clearer manner.
- for the statistical significance, authors adopted a letter code that should be better explained, as it results hard to understand.
- I'm not familiar with the "ATP turnover" parameter. As it is described in M&M section, it seems to me the classic "ATP production" one; yet, in the figure descriptions authors mention both ATP production and ATP turnover, while failing to show data for ATP production. Please explain and correct this issue.
- line 167: there is no figure 2c-d, even if it would be interesting to see data of the 24h recovery from 10uM dox. Are Authors referring to 3c-d? in such case, as I stated above there is a huge difference with the control condition, it is very difficult to see the protective effect authors are trying to demonstrate.
- line 199-201: Authors have no proof that the low spare respiratory capacity of the BPF condition is due to the high basal respiration and not to a specific effect of the BPF, especially in light of data they showed with the 200uM BPF concentration. Please eliminate this statement.
Author Response
Response to reviews
We are very grateful for the reviews provided by the Reviewers. The comments are encouraging and the comments appear to share our judgement that this study and its results are scintifically important. Please see below, in blue, our detailed response to Reviewers’ suggestions. All revisions refer to the manuscript file with tracked changes.
Reviewer #1
In this work, authors try to demonstrate a protective effect of bergamot polyphenolic fraction (BPF) from the cardiotoxic side-effects of doxorubricin (Dox) on porcine aortic endothelial cells.
They investigate the morphology and the bioenergetic parameters of this cells exposed or not to Dox and BPF.
Overall, while the manuscript is generally well written and quite clear, however there are some concerns that prevent its publication in the current form.
The main issue is that 200uM BPF appears to have an impact on cellular metabolism alone, as they inhibits both maximal and spare respiratory capacity in figure 2a-b, as well as basal, maximal and "ATP turnover" in figure 3a-b and 3c-d. The latter figure (3c) shows a rather alarming profile, with a curve for BPF alone that is more similar to the Dox one than to the control. This data seems to suggest a possible detrimental effect rather than a protective one. Indeed, I'd like to see the results on cell viability of BPF alone, as figure 1F shows only data in combination with Dox. Conversely, a better adherence to authors' hypothesis is shown by 100uM BPF (fig 4), in which the protective effects appears to be more convincing. Did Authors investigated the effects of lower concentrations, such as 50uM BPF?
We thank the reviewer for bringing up this important issue. We have started the experiments by using 200 ug/mL BPF considering the effect obtained in our pilot work (PMID: 35563707). Indeed, 200 ug/mL BPF (5h treatment) increased the MTT value, whereas the optimum cell metabolism activity was observed with 100 ug/mL BPF. In the present work, we observed a time- and concentration-dependent reduction of pAECs bioenergetic parameters in presence of 200 ug/mL BPF at 24h (Fig. 2 and Fig. 3A,B) and a little more accentuated at 48h (Fig. 3C,D). However, the MTT test of pAECs was not affected 200 ug/mL BPF (see results in the Figure below). This sentence has been added to the text.
The Reviewer asserts that “The latter figure (3c) shows a rather alarming profile, with a curve for BPF alone that is more similar to the Dox one than to the control”. On the contrary, we can see that curve for BPF alone is more similar to the Dox plus BPF. Therefore, we have noticed an improvement in cell respiration if compared to Dox-treated pAECs. 100 ug/mL BPF, in the same way as our paper PMID: 35563707, was the concentration that performed a protective effect of pAECs cell metabolism on the dose-dependent and time-dependent Dox toxicity. We think that the protective effect of BPF is on permeability transition pore (PTP) opening triggered by Dox. 100 ug/mL BPF inhibited the PTP, whereas 50 ug/mL BPF had a low desensitizing effect on the PTP formation (PMID: 35563707). Consequently, we have set concentrations higher than 100 ug/mL BPF to counteract the toxic effect of Dox.
In the introduction and discussion sections, authors state that Dox exerts its action thorugh ROS production. Since ROS are easily measurable with flow cytometry, Authors should consider to include these experiments to strenghten their results and demonstrate that BPF causes ROS reduction.
Thank you for this suggestion. It would have been interesting to explore this aspect. However, in our study, this would not be possible because this paper is only a “Communication”. Indeed, this aspect has been speculated as a possible cause of PTP formation by Dox toxicity. However, we will consider this suggestion in future work, thanks.
Moreover, it is unclear how they performed normalization for the XF analyzer data, cell were counted at the seeding stage, before or after the analysis? Please provide an extended explanation, as if performed at the seeding stage with the 50% of dox mortality it would invalidate the results
We are delighted with the Reviewer's careful review of our work. This comment is tangible proof!
However, according to the reviewer’s comment, the WAVE program of Seahorse XFp has been set to normalize the result based on pAECs vitality obtained from the MTT test. Thanks.
minor points
- In the abstract, line 24-26, authors would mean that mitochondrial parameter are impaired or improved? Please reformulate the phrase in a clearer manner.
Done, thanks. The parameters were improved.
- for the statistical significance, authors adopted a letter code that should be better explained, as it results hard to understand.
Thank you for pointing out this deficiency. We have edited the captions of the figures.
- I'm not familiar with the "ATP turnover" parameter. As it is described in M&M section, it seems to me the classic "ATP production" one; yet, in the figure descriptions authors mention both ATP production and ATP turnover, while failing to show data for ATP production. Please explain and correct this issue.
We thank the reviewer for bringing up this important issue. According to the reviewer’s comment, we have changed “ATP production” in “ATP turnover” in the caption in the same way as was reported in the figures. We agree with the Reviewer that the ATP turnover consists of all reactions involved in phosphorylation of ADP to ATP and the export of ATP in the extramitochondrial space.
- line 167: there is no figure 2c-d, even if it would be interesting to see data of the 24h recovery from 10uM dox. Are Authors referring to 3c-d? in such case, as I stated above there is a huge difference with the control condition, it is very difficult to see the protective effect authors are trying to demonstrate.
We would like to thank the reviewer for pointing out our error. We have arranged the text highlighting the restoration of cellular respiratory activity only compared to DOX treatment (not control) when pAECs are treated with 200 μg/ml BPF.
- line 199-201: Authors have no proof that the low spare respiratory capacity of the BPF condition is due to the high basal respiration and not to a specific effect of the BPF, especially in light of data they showed with the 200uM BPF concentration. Please eliminate this statement.
Done, thanks

Reviewer 2 Report
The topic of the presented manuscript is very interesting. has an application and cognitive character. However, as a reviewer, I have to point out a few minor issues:
1. I recommend that you should standardize the units either micromoles or micrograms
2. Figure 1F is unreadable, I recommend adding it as a separate figure
3. I suggest to explain abbreviations e.g. FBS
4. Use for doxorubicin capital letters – DOX, then the text will be more readable
Author Response
Response to reviews
We are very grateful for the reviews provided by the Reviewers. The comments are encouraging and the comments appear to share our judgement that this study and its results are scintifically important. Please see below, in blue, our detailed response to Reviewers’ suggestions. All revisions refer to the manuscript file with tracked changes.
Reviewer #2
The topic of the presented manuscript is very interesting. has an application and cognitive character. However, as a reviewer, I have to point out a few minor issues:
- I recommend that you should standardize the units either micromoles or micrograms
BPF is a phytocomplex of unknown molecular mass that does not allow the calculation of moles. For this reason, we have used the μg/mL unit for the BPF.
- Figure 1F is unreadable, I recommend adding it as a separate figure
This is a Communication and we have to use a low number of figures. However, we have increased the resolution and size of Figure 1. Thank you for pointing out this deficiency.
- I suggest to explain abbreviations e.g. FBS
Done, thanks.
- Use for doxorubicin capital letters – DOX, then the text will be more readablee
Done, thanks.
Round 2
Reviewer 1 Report
I'm rather satisfied with the answer authors provided, but I recommend the following minor correction:
- Maybe I wasn't clear enough in the last round of revision, but I meant that I've never found the "ATP Turnover" parameter in other works. The description given by the Authors in the M&M section corresponds to the "ATP Production" parameter. The XF community has always called it the "ATP Production". So, please change "ATP Turnover" in "ATP Production". Accordingly, please correct parameter list in lines 162-163 as there is one in excess.
- Please add in the M&M how normalization was applied and mortality taken into account as Authors explained to me in the reply.
Author Response
Response to review
Thank you for giving us the opportunity to submit a second revised draft of the manuscript “The impairment of cell metabolism by cardiovascular toxicity of doxorubicin is reversed by bergamot polyphenolic fraction treatment in endothelial cells” for publication in the International Journal of Molecular Sciences. We appreciate the time and effort that the reviewer dedicated to providing feedback on our manuscript and we are grateful for the insightful comments and valuable improvements to our paper. We addressed the points raised by the Reviewer as far as possible and detailed responses to the Reviewer’s comments are given below. We heartily hope that, in the revised version, our manuscript might be accepted for publication.
Reviewer #1
I'm rather satisfied with the answer authors provided, but I recommend the following minor correction:
- Maybe I wasn't clear enough in the last round of revision, but I meant that I've never found the "ATP Turnover" parameter in other works. The description given by the Authors in the M&M section corresponds to the "ATP Production" parameter. The XF community has always called it the "ATP Production". So, please change "ATP Turnover" in "ATP Production". Accordingly, please correct parameter list in lines 162-163 as there is one in excess.
Thank you for pointing out this deficiency. We agree with the Reviewer and revised the manuscript accordingly.
- Please add in the M&M how normalization was applied and mortality taken into account as Authors explained to me in the reply.
We have edited the M&M accordingly, thanks.